# In-Vitro Assessment of the Corrosion Potential of an Oral Strain of Sulfate-Reducing Bacteria on Metallic Orthodontic Materials

**DOI:** 10.3390/ijerph192215312

**Published:** 2022-11-19

**Authors:** Umarevathi Gopalakrishnan, Kavitha Thiagarajan, A. Sumathi Felicita, Pallabhi Gosh, Abdulrahman Alshehri, Wael Awadh, Khalid J. Alzahrani, Fuad M. Alzahrani, Khalaf F. Alsharif, Ibrahim F. Halawani, Saleh Alshammeri, Ahmed Alamoudi, Dhalia H. Albar, Hosam Ali Baeshen, Shankargouda Patil

**Affiliations:** 1Department of Orthodontics, Sri Venkateswara Dental College and Hospital, Thalambur, Chennai 600130, India; 2Department of Dental Surgery, Government Stanley Medical College and Hospital, Chennai 600001, India; 3Department of Orthodontics, Saveetha Dental College and Hospital, Saveetha Institute of Medical and Technical Sciences, Chennai 600077, India; 4Biomedical Engineer, Saveetha Dental College and Hospital, Chennai 600077, India; 5Division of Orthodontics, Department of Preventive Dental Sciences, Faculty of Dentistry, Jazan University, Jazan 45142, Saudi Arabia; 6Department of Clinical Laboratories Sciences, College of Applied Medical Sciences, Taif University, Taif 21944, Saudi Arabia; 7Department of Optometry, College of Applied Medical Sciences, Qassim University, Buraydah 1162, Saudi Arabia; 8Oral Biology Department, Faculty of Dentistry, King Abdulaziz University, Jeddah 22254, Saudi Arabia; 9Department of Preventive Dental Sciences, College of Dentistry, Jazan University, Jazan 45142, Saudi Arabia; 10Department of Orthodontics, Faculty of Dentistry, King Abdulaziz University, Jeddah 21589, Saudi Arabia; 11College of Dental Medicine, Roseman University of Health Sciences, South Jordan, UT 84095, USA; 12Centre of Molecular Medicine and Diagnostics (COMManD), Saveetha Dental College & Hospitals, Saveetha Institute of Medical and Technical Sciences, Saveetha University, Chennai 600077, India

**Keywords:** archwires, corrosion, metallic orthodontic brackets, mini implants, sulfate reducing bacteria

## Abstract

Aim: Orthodontic literature is scant when it comes to microbial corrosion. The oral prevalence of many bacteria which are capable of causing microbial corrosion is reported in the dental literature. The aim of this study is to experimentally determine the corrosive potential of an oral strain of Sulfate-reducing bacteria. Materials and Methods: Stainless steel (SS) bracket, stainless steel archwire, NiTi archwire, Titanium molybdenum (TMA) archwire, and titanium miniscrew were immersed in five media which included Artificial saliva (group I), Sulfate rich artificial saliva (group II), API agar medium specific for SRB (group III), AS + API medium+ bacterial strain (group IV), SRAS+ API medium+ bacterial strain (group V). The materials were then subjected to Scanning electron microscopy and energy-dispersive X-ray analysis (EDX). Results: Materials in groups I, II, and III did not show any surface changes whereas materials in groups IV and V which contained the bacteria showed surface changes which were erosive patches suggestive of corrosion. EDX analyses were in line with similar findings. Conclusion: This in vitro study suggested that the oral strain of Sulfate-reducing bacteria was able to induce corrosive changes in the experimental setup.

## 1. Introduction

Corrosion in orthodontics is a hugely researched topic since the effect of corrosion affects the biomechanical properties of the metallic materials used in orthodontic treatment. The most significant effect is increased friction which often reduces the effective force delivered while the other effects are longer treatment time, leakage of corrosion products, etc. Despite this, the orthodontic literature is inadequate when it comes to research in Microbially induced corrosion (MIC). MIC is not any different from other types of corrosion except that the organisms contribute to the electrochemical process as a by-product of their metabolism which includes the production of corrosive acids like carboxylic acid, production of hydrogen sulfide, etc. [1,2] Microorganisms that have been attributed to MIC include some bacteria like sulfate-reducing bacteria, fungi like *cladosporium resinae* and algae like *Nostoc parmelioides* to name a few. The organism most commonly implicated in MIC are the ones involved in sulfur cycles, especially the sulfate-reducing bacteria (SRB) [2]. In the dental literature, the importance of microbial corrosion has been scantly addressed which is evident from a recent bibliometric analysis [3]. These organisms which can cause microbial corrosion are actively involved in the initiation and acceleration of metal dissolution. In one particular literature, SRB’s corrosion potential was being used for dissolving the fractured files in root canals [4]. Since intraoral corrosion leaks out corrosive products like nickel ions in the oral cavity [5] and moreover alters the biomechanical properties of the metallic orthodontic materials [6] and SRB is prevalent in the oral cavity of even healthy patients [7,8], it is not only inquisitive but also imperative to study whether SRB in the oral cavity was capable of causing corrosion of metallic orthodontic materials. Hence the aim of this in vitro study is to experimentally determine the corrosion potential of a particular strain of SRB from the oral cavity, on metallic orthodontic materials.

## 2. Materials and Methods

### 2.1. Sample Selection

The strain of SRB *Desulfovibrio* isolated from the oral cavity of an orthodontic patient from a previous study by the author [9] was used as the strain for this in vitro study. In this study [6], DNA amplification based on universal 16S rRNA and subsequent sequencing confirmed the presence of SRB specific to *D. desulfuricans*, *D. piger*, and *D. fairfieldensis* based on the identification in the existing database at GenBank. The obtained sequences were submitted to the NCBI GenBank repository, which can be accessed using the accession numbers ON183261, ON183262, and ON183263. Samples of metal orthodontic brackets (3M^TM^ unitek Victory series^TM^, St. Paul, MN, USA), NiTi archwire (Unitek™ Nitinol Classic Archwire), Stainless steel archwire (G&H orthodontics, Franklin, IN, USA), Titanium molybdenum alloy (TitanMoly™) and Titanium Mini implant (Ortho one Inc., Coimbatore, India) were included for study and sent to the lab for elemental analysis by Energy-dispersive X-ray spectroscopy.

### 2.2. Assembly of Samples

Each sample was tested in five media: (a) Artificial saliva (AS) (group I), (b) Sulfate rich artificial saliva (SRAS) (group II), (c) API agar medium specific for SRB (13005 Sulfate API Agar, Sigma Aldrich, St. Louis, MO, USA) (group III), (d) AS + API medium+ bacterial strain (group IV) (e) SRAS+ API medium+ bacterial strain (group V). The pH of the AS, SRAS, and API medium was adjusted to 7.4 ± 0.2. The first three media served as the control to rule out corrosion caused by the saliva or the API medium. Artificial saliva and sulfate-rich artificial saliva were prepared in accordance with Heggendorn et al. [10] wherein Sulfate rich AS is prepared by increasing the Na_2_SO_4_ from 0.5832/L to 1.0/L to provide a favorable environment for the growth of SRB. The blackening of the medium indicated the formation of iron sulfide by the bacteria, and this was taken as positive for the identification of bacterial growth. The five media were kept in 1.5 mL Cryo vials for immersion tests. The test samples including the brackets, archwires (NiTi, SS, and TMA), and the mini-implant were polished and autoclaved at 121 °C for 20 min. They were examined under a Scanning electron microscope before immersing them in the cryo vials with the medium. They were immersed in the corresponding medium at 37 °C for 60 days after which they were taken out for assessment of corrosion.

### 2.3. Assessment of Corrosion Potential

After 60 days of immersion, the samples were subjected to scanning electron microscopy (JEOL JSM IT800 SHL SCANNING ELECTRON MICROSCOPE, JEOL, Inc. Pleasanton, CA, USA) and EDX analysis to compare the before and after immersion images and their composition, respectively. The Energy Dispersive X-ray Analysis (EDX) is an X-ray technique to identify the metallic composition, bacterial and corrosion products in relation to spatial distribution.

## 3. Results

### 3.1. Evaluation of Corrosion Potential of Artificial Saliva and Culture Medium

A comparison of the images of the samples before and after immersion in the artificial saliva, sulfate-rich saliva, and SRB-specific culture medium did not show any presence of corrosion in the form of surface roughness, pitting, or surface deposits.

### 3.2. Evaluation of the Corrosive Potential of the Genus Desulfovibrio

Examination of the images of the samples through SEM from groups IV and V revealed SRB colonies growing as discrete nodules on the surface of the wire under low magnification (Figure 1A), the same nodules revealed circular patches with linear patterns of bacterial growth within the circular patches under higher magnification (Figure 1B). There were patches of erosion and corrosion pits seen on the surface of the samples from both groups IV and V for all the materials tested viz, stainless steel brackets (Figure 2), NiTi wire (Figure 3A), stainless steel wire (Figure 3B), TMA wire (Figure 3C) and mini-implant (Figure 3D).

### 3.3. EDX Analysis of the Samples before and after Immersion in Media with SRB

EDX spot analysis was done before and after immersion in selected spectrums. Post immersion analysis was done in areas involving surface roughness, irregularity, nodular growth, and pitting. Post immersion increases in the weight percentage of C, S, and O ions were seen in stainless steel brackets, stainless steel, TMA, and NiTi wires (Figure 4). The S and O ions may be suggestive of corrosion activities of SRB while the C ions are suggestive organic biofilm and bacterial components. Nodular areas which were suggestive of bacterial growth had greater content of C, Ca, O, and P suggestive of organic bacterial content (Figure 5).

## 4. Discussion

Corrosion in a neutral scenario generally begins with the formation of corrosive cells. The five key elements in corrosion are the anode, cathode, electrolyte, electrical connection, and potential difference. In the case of dissimilar metals, this corrosive cell is formed due to the electric potential difference between the metals. In the case of similar metal, the corrosive cell forms due to the potential difference between different areas/points due to different concentrations of reactants. Intra orally the situation becomes even more favorable attributed to saliva which can serve as a good electrolyte [11] as well as due to biofilms. Biofilm is a surface film composed of organic and inorganic saliva components that are colonized with microorganisms in extracellular polymeric substances adsorbed on all surfaces in the oral cavity including the metallic appliances [12]. Metallic surfaces covered by biofilm are protected from outside oxygen contact whilst metallic surfaces free from biofilm have access to oxygen outside. This difference in oxygenation creates a corrosive cell and the metal under the biofilm becomes the anodic site leaching electrons and thereby corroding. The electrons from the anodic site reach the metal surface freely accessible to oxygen where it forms hydroxyl ions. This is the mechanism by which generally microorganisms contribute to corrosion through the formation of biofilms and oxygen concentration corrosive cells^2^. In Microbially induced corrosion (MIC), the organism causes corrosion as a result of their metabolic process rather than with the simple physical hindrance causing oxygenation differences with biofilms. This sort of microbially induced corrosion was first reported in 1934 when Von Wolzogen Kuhr and Van Der Vlugt proposed the microbiological theory for the underground corrosion of iron pipelines in the absence of stray electric currents [13]. The most important bacteria which cause this microbial corrosion are the sulfate-reducing bacteria which have the ability to conduct sulfate reduction as part of their metabolic dissimilation. Though they are strict anaerobes they can survive the aerobic environment for a longer time. They have the unique property of conducting dissimilatory sulfate reduction wherein they reduce sulfate by using it as the terminal electron receptor converting it to sulfide.

The biofilm formation on the metallic surface is the initial step in microbial corrosion [13] by SRB. Numerous bacteria along with SRB get adhered to the biofilm formed on the metallic surface. SRB uses quorum sensing to optimize the adhesion to biofilms. When compared to non-microbial corrosion, corrosion induced by SRB or other similar bacteria will usually be localized and pitted since bacteria gets adhered to that specific site and acts specifically from that site. The metallic atoms that get released from the anodic site combine with the sulfide made available by the dissimilatory reduction of SRB to form metallic sulfide. The electrons from the anodic site bind with ionized hydrogen from an aqueous environment to form hydrogen gas. This is the rate-limiting step in the corrosion of any metallic elements. SRB accelerates this process by consuming the hydrogen gas through their hydrogenase enzyme and thus depolarizing the cathodic site. Iron was the most studied metal for microbial corrosion but metallic ions other than iron also undergo a similar oxidation-reduction process by SRB [14]. There is another mechanism suggested for the corrosion induced by SRB via the production of hydrogen sulfide during biological sulfate reduction. Hydrogen sulfide is a known chemical corrosive agent. This is known as chemical microbially influenced corrosion (CMIC) [15]. Yet another suggested mechanism is the secretion of extracellular polymer substances by the SRB which are capable of binding to metal ions. This chelating property has also been suggested to accelerate microbial corrosion by SRB [16].

A 60-day immersion period was chosen for the study based on the reports of Isa et al. [17] that the SRB in anaerobic reactors were most active between 11 and 24 days after which the activity decreased with a reduction in sulfide production after 50 days as a supersaturation with FeS2 will inhibit the further corrosion of steel [18]. The corrosive property of SRB has been experimentally elicited by other authors like Heggendorn et al. [4,19] who showed the biofilm interaction with the metallic surface of endodontic files and the consequent corrosion. The results of our study are in line with their study wherein we identified localized corrosion induced by SRB of genus *Desulfovibrio* in stainless steel brackets and wires, NiTi, TMA wires, and Titanium mini-implants. The increased prevalence of SRB in orthodontic patients has already been studied by the same author in previous work. With this increased prevalence and the experimentally proven corrosive potential in various metals used in orthodontic practice, the possibility of the SRB, especially the genus *Desulfovibrio,* inducing microbial corrosion in clinical scenarios is very likely. In fact, microbial corrosion caused by *Desulfovibrio* has been tested as a means to dissolute endodontic files that have been left fractured in the canal by Heggendorn et al. [10].

## 5. Conclusions

The study gives an experimental insight into the corrosion caused by the *Desulfovibrio* strain of SRB that was isolated from the oral cavity of orthodontic patients. This study indicates that bacteria that are prevalent in the oral cavity have the potential to corrode metals that are used in orthodontic treatment. This study has a greater clinical significance given the potential effects of corrosion in the biomechanical alteration of the metallic properties of orthodontic materials, as well as the cytotoxic effects of the corrosive products. It has to be extended to studying the various other strains that are prevalent in the oral cavity which could have the ability to corrode other metallic elements like implants, prosthetic materials, etc. Another future perspective is to find antibacterial solutions to address the corrosion caused by these bacteria.

## Figures and Tables

**Figure 1 ijerph-19-15312-f001:**
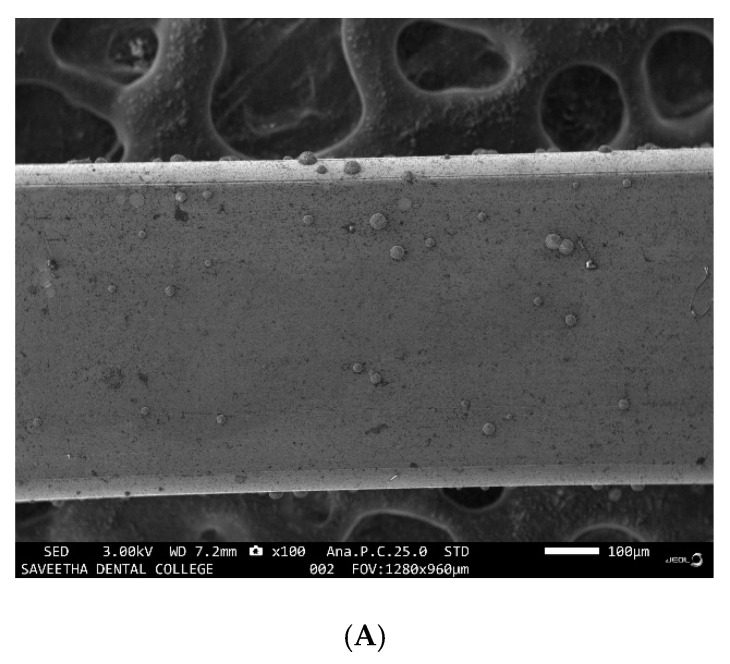
SRB growth on the surface of TMA wire by SEM under (**A**) lower magnification of 100× and (**B**) high magnification of 250×.

**Figure 2 ijerph-19-15312-f002:**
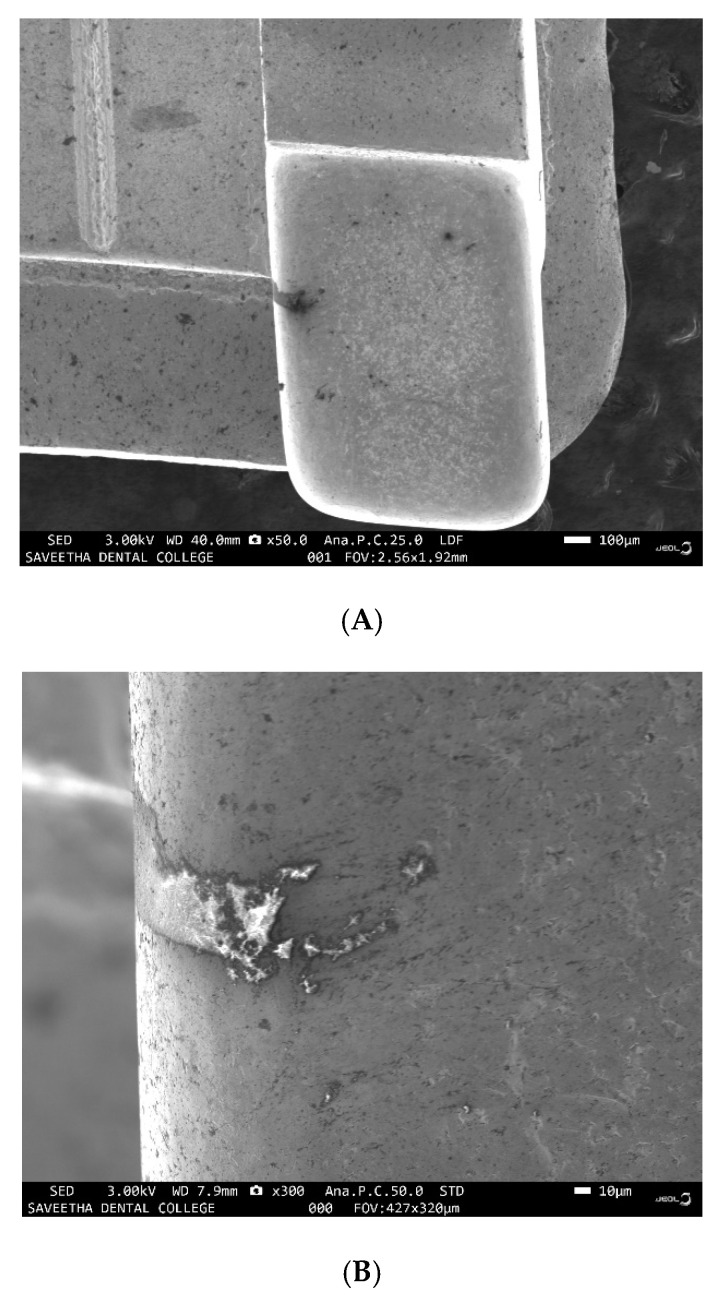
Corrosion patch seen on surface of SS bracket under (**A**) low 50× and (**B**) high magnification 300×.

**Figure 3 ijerph-19-15312-f003:**
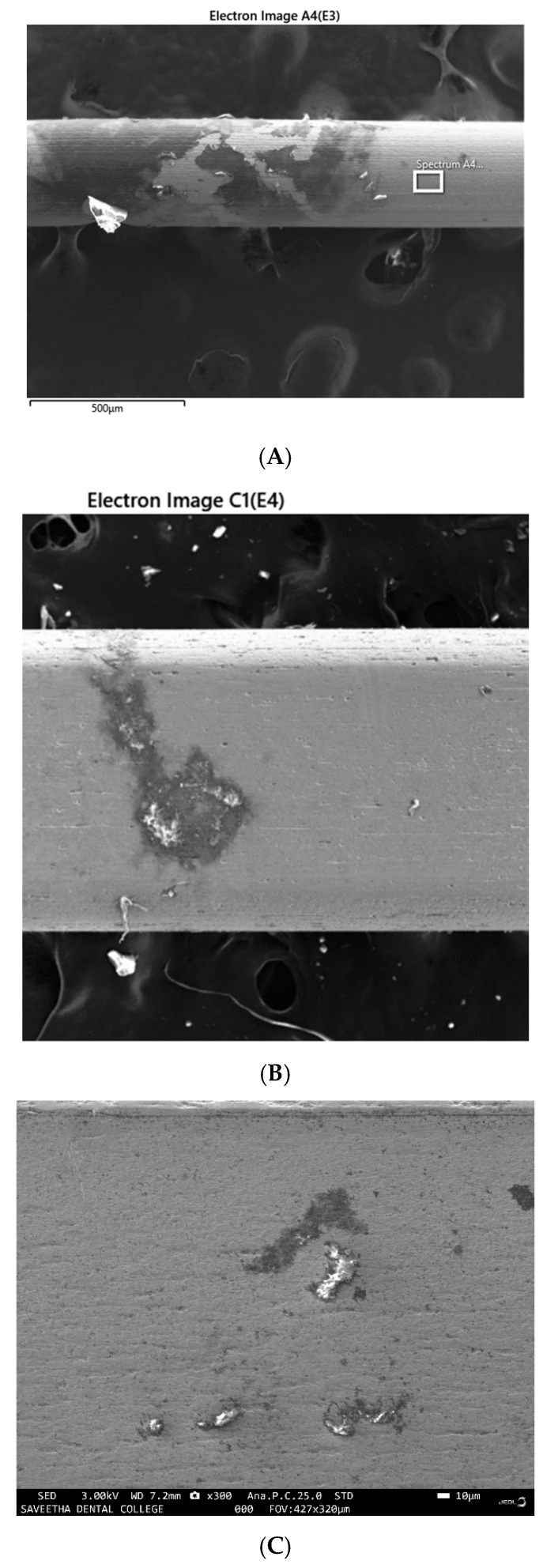
Corrosion patches observed on various archwires (**A**) 0.016 NiTi wire (**B**) SS wire (**C**) TMA wire (**D**) Mini implant Titanium screw.

**Figure 4 ijerph-19-15312-f004:**
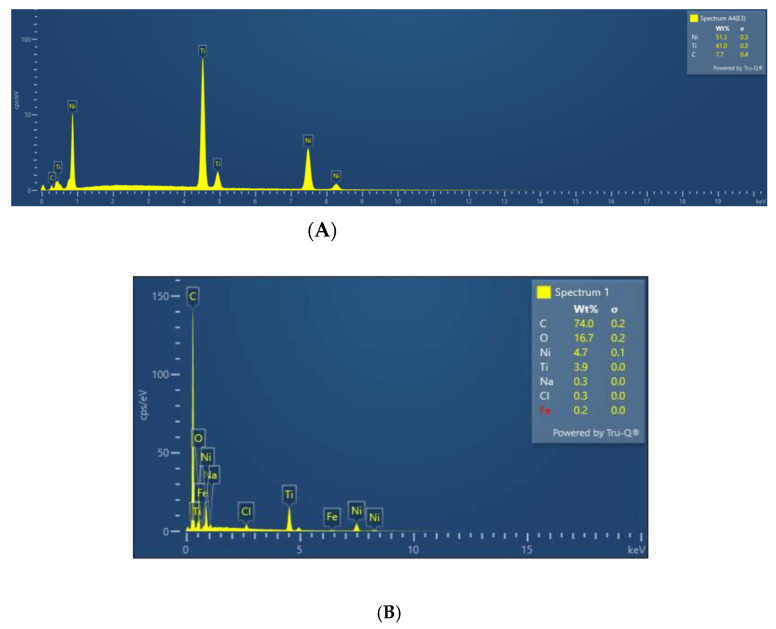
(**A**) NiTi before immersion; (**B**) NiTi after immersion; (**C**) Stainless steel before immersion; (**D**) Stainless steel after immersion.

**Figure 5 ijerph-19-15312-f005:**
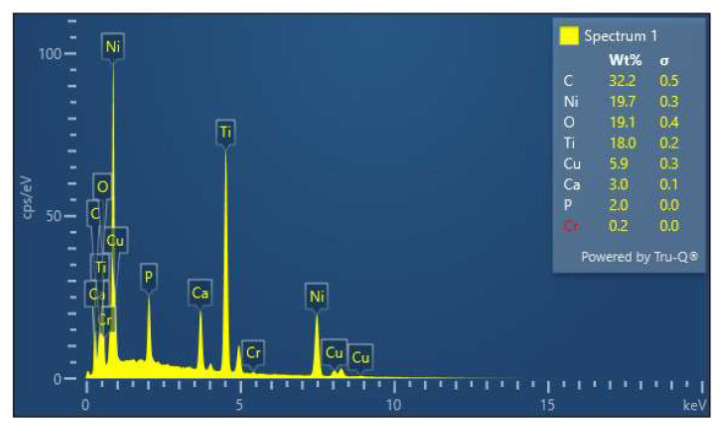
EDX of the nodular growth on surface of NiTi wire.

## Data Availability

Not applicable.

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
