# Peer review of "In-Vitro Assessment of the Corrosion Potential of an Oral Strain of Sulfate-Reducing Bacteria on Metallic Orthodontic Materials"

_ijerph, 2022, doi:10.3390/ijerph192215312_

Round 1

Reviewer 1 Report

Thank you for the opportunity to review, I have the following comments.

Introduction

1.     In my opinion, the introduction is too short. It should be elaborated on to give a better idea of the subject matter of the study.

2.     L56 – ‘’Corrosion in orthodontics is a hugely researched topic.’’ -  Explain why corrosion in orthodontics is such an important and researched topic. Your article will also be read by people not related to the topic of orthodontics. Therefore, bring the subject closer.

3.     L59 – ‘’ electrochemical process as a by-product of their metabolism’’ - describe this proces.

4.     L59-60 – ‘’ Microorganisms that have been attributed to MIC include some bacteria, fungi and algae.’’ - Podaj przykÅ‚ady z nazwy niektórych bakterii, grzybów i glonów oraz dokÅ‚adne cytaty na ten temat.

5.     L62-63 –‘’ In the dental literature, the importance of microbial corrosion has been scantly addressed.’’ - This is a very subjective opinion. Why do the authors think so? Did they find support from other studies ? If so please cite or explain further their assumptions.

6.     L64-65 – ‘’ In a recent literature’’ - In my opinion, the 2018 publication cannot be called the latest literaturÄ™.

7.     L68 ‘’ causing corrosion of metallic orthodontic materials.’’ -  Why it would be important, what health and economic benefits would be associated with it . Expand on this thought.

8.     You should also add an explicit justification of the research topic in the introduction.

Materials and Methods

9.     L74- 75 -  ‘’ The strain of SRB Desulfovibrio isolated from the oral cavity of an orthodontic patient from a previous study by the author[6]’’ – The authors refer to a study:

10.  Sulfate-Reducing Bacteria in Patients Undergoing Fixed Orthodontic Treatment. Int. Dent. J. 2022, doi:10.1016/j.identj.2022.07.007

11.  In the earlier study, the authors cite that they collected material in accordance with the Declaration of Helsinki and the Bioethics Commission. I think that after although it is the same material they should write all the statements as in the earlier study and cite the bioethical approval number in this study in the section on statements (L217).

. Results

12.  As it stands, figures 4 and 5 are unreadable. I am unable to evaluate them. I suggest adding them in better quality.

Conclusions

13.  A clear sentence summarizing the results is missing.

Editorial errors

14.  There should be a space between the end of the sentence and the quote - the remark applies to the entire text.

15.  After the quote, put a period at the end of the sentence  - the remark applies to the entire text.

16.  The text should be justified - the remark applies to the entire text.

17.  L159 – ‘’ first reported in 1934 when Von Wolzogen Kuhr and Van Der’’ - authors refer to a paper from 1934 however they refer to a paper from 2009 - suggests referring to the work of Von Wolzogen Kuhr and Van De directly.

18.  I understand that the topic is niche however, the authors mostly refer to outdated research. 7 papers are older than 22 years. I think that the literature should be enriched with newer publications.

Author Response

Reviewer 1 corrections are marked in red

Reviewer 2 corrections are marked in yellow

Reviewer 1:

Thank you for the opportunity to review, I have the following comments.

Introduction

  1. In my opinion, the introduction is too short. It should be elaborated on to give a better idea of the subject matter of the study.

     Response to Reviewer:  The introduction has been elaborated as per the suggestion

  1. L56 – ‘’Corrosion in orthodontics is a hugely researched topic.’’ -  Explain why corrosion in orthodontics is such an important and researched topic. Your article will also be read by people not related to the topic of orthodontics. Therefore, bring the subject closer.

     Response to Reviewer:  Explanation added

  1. L59 – ‘’ electrochemical process as a by-product of their metabolism’’ - describe this proces.

     Response to Reviewer:  Explanation added

  1. L59-60 – ‘’ Microorganisms that have been attributed to MIC include some bacteria, fungi and algae.’’ - Podaj przykÅ‚ady z nazwy niektórych bakterii, grzybów i glonów oraz dokÅ‚adne cytaty na ten temat.

    Response to Reviewer:   Examples given

  1. L62-63 –‘’ In the dental literature, the importance of microbial corrosion has been scantly addressed.’’ - This is a very subjective opinion. Why do the authors think so? Did they find support from other studies ? If so please cite or explain further their assumptions.

      Response to Reviewer:  Explanation added

  1. L64-65 – ‘’ In a recent literature’’ - In my opinion, the 2018 publication cannot be called the latest literature.

       Response to Reviewer:  Corrected

  1. L68 ‘’ causing corrosion of metallic orthodontic materials.’’ -  Why it would be important, what health and economic benefits would be associated with it . Expand on this thought.

      Response to Reviewer:  Explanation added

  1. You should also add an explicit justification of the research topic in the introduction.

      Response to Reviewer:  Explanation added

Materials and Methods

  1. L74- 75 -  ‘’ The strain of SRB Desulfovibrio isolated from the oral cavity of an orthodontic patient from a previous study by the author[6]’’ – The authors refer to a study:

       Response to Reviewer: Corrections done

  1. Sulfate-Reducing Bacteria in Patients Undergoing Fixed Orthodontic Treatment. Int. Dent. J. 2022, doi:10.1016/j.identj.2022.07.007

     Response to Reviewer: Corrections done

  1. In the earlier study, the authors cite that they collected material in accordance with the Declaration of Helsinki and the Bioethics Commission. I think that after although it is the same material they should write all the statements as in the earlier study and cite the bioethical approval number in this study in the section on statements (L217).

       Response to Reviewer: Corrections done

Results

  1. As it stands, figures 4 and 5 are unreadable. I am unable to evaluate them. I suggest adding them in better quality.

     Response to Reviewer: Legible pictures added

Conclusions

  1. A clear sentence summarizing the results is missing.

     Response to Reviewer: Correction done as per suggestion

Editorial errors

  1. There should be a space between the end of the sentence and the quote - the remark applies to the entire text.

      Response to Reviewer: Corrections done

  1. After the quote, put a period at the end of the sentence  - the remark applies to the entire text.

       Response to Reviewer: Corrections done

  1. The text should be justified - the remark applies to the entire text.

       Response to Reviewer: Corrections done

  1. L159 – ‘’ first reported in 1934 when Von Wolzogen Kuhr and Van Der’’ - authors refer to a paper from 1934 however they refer to a paper from 2009 - suggests referring to the work of Von Wolzogen Kuhr and Van De directly.

      Response to Reviewer: Correction done

  1. I understand that the topic is niche however, the authors mostly refer to outdated research. 7 papers are older than 22 years. I think that the literature should be enriched with newer publications.

       Response to Reviewer: Corrections done

Reviewer 2:

This is an interesting manuscript covering an important interaction in the oral cavity by sulfate reducing bacteria. There are a few places where more information is needed for the readers or some corrections could be made.

Abstract

Line 41; Should read 'five media' not 'five mediums'

Response to Reviewer: Corrections done

Line 42; change 'includes' to 'included'

Response to Reviewer: Corrections done

Line 43; add a comma between AS+API medium + bacterial strain and SRAS (e.g., strain, SRAS) Response to Reviewer: Corrections done

Introduction

Line 66; change to 'SRB in the oral cavity

Response to Reviewer: Corrections done

Lines 67 and 68; should either read 'whether a SRB from the oral cavity was capable' or 'whether SRB from the oral cavity were capable'

Response to Reviewer: Corrections done

Line 69; change 'invitro' to 'in vitro'

Response to Reviewer: Corrections done

Line 70; change to 'from the oral cavity'

Response to Reviewer: Corrections done

Materials and Methods

Line 74; change to 'The SRB was a strain of Desulfovibrio'. Italicize the genus name. Because this strain was previously isolated and identified, can the authors indicate for the readers how the the bacterium was identified? Was it through 16S rRNA sequencing or phenotypic tests?

Response to Reviewer: Corrections done

Line 75: change to 'in vitro'

Response to Reviewer: Corrections done

Line 82: change to 'five media'

Response to Reviewer: Corrections done

Line 84; change to 'Sigma Aldrich'

Response to Reviewer: Corrections done

Line 86; change to 'The first three media'

Response to Reviewer: Corrections done

Some more methodology would be valuable. How were the media inoculated with the bacterium? How was the growth of the bacterium in the media assesses? Was growth of the bacterium the same in both inoculated media? This is impotant as it could have influenced the results.

Response to Reviewer: corrections included

Assessment of corrosion potential

The authors state that after 60 days of immersion the samples were chemically stripped to remove the corrosion impurities. How was the chemical stripping accomplished? And with what? This procedure would also suggest that any attached bacteria and/or biofilms would be removed by the stripping. Was that the case? If so, how were Figures 1A and B accoplished? Were separate samples with biofilms/attached bacteria that were not chemically stripped used for those figures? If so, methods of how this was accomplished should be presented. In any case, more methodology needs to be inserted here.

Response to Reviewer: Thank you for pointing this out error. Chemical stripping was not done before SEM analysis. Corrections done intext.

References

Some of the references appear to be incomplete in terms of authorship. The authors should carefully go through the list of references and make sure they all correspond to instructions given by the journal.

Response to Reviewer: Corrections done

Reviewer 2 Report

This is an interesting manuscript covering an important interaction in the oral cavity by sulfate reducing bacteria. There are a few places where more information is needed for the readers or some corrections could be made.

Abstract

Line 41; Should read 'five media' not 'five mediums'

Line 42; change 'includes' to 'included'

Line 43; add a comma between AS+API medium + bacterial strain and SRAS (e.g., strain, SRAS)

Introduction

Line 66; change to 'SRB in the oral cavity'

Lines 67 and 68; should either read 'whether a SRB from the oral cavity was capable' or 'whether SRB from the oral cavity were capable'

Line 69; change 'invitro' to 'in vitro'

Line 70; change to 'from the oral cavity'

Materials and Methods

Line 74; change to 'The SRB was a strain of Desulfovibrio'. Italicize the genus name. Because this strain was previously isolated and identified, can the authors indicate for the readers how the the bacterium was identified? Was it through 16S rRNA sequencing or phenotypic tests?

Line 75: change to 'in vitro'

Line 82: change to 'five media'

Line 84; change to 'Sigma Aldrich'

Line 86; change to 'The first three media'

Some more methodology would be valuable. How were the media inoculated with the bacterium? How was the growth of the bacterium in the media assesses? Was growth of the bacterium the same in both inoculated media? This is impotant as it could have influenced the results.

Assessment of corrosion potential

The authors state that after 60 days of immersion the samples were chemically stripped to remove the corrosion impurities. How was the chemical stripping accomplished? And with what? This procedure would also suggest that any attached bacteria and/or biofilms would be removed by the stripping. Was that the case? If so, how were Figures 1A and B accoplished? Were separate samples with biofilms/attached bacteria that were not chemically stripped used for those figures? If so, methods of how this was accomplished should be presented. In any case, more methodology needs to be inserted here.

References

Some of the references appear to be incomplete in terms of the authorship. The authors should carefully go through the list of references and make sure they all correspond to instructions given by the journal.

Author Response

(The authors gave the same response as above.)

Round 2

Reviewer 1 Report

The authors' responses are acceptable.